# Interaction between Acute Hepatic Injury and Early Coagulation Dysfunction on Mortality in Patients with Acute Myocardial Infarction

**DOI:** 10.3390/jcm12041534

**Published:** 2023-02-15

**Authors:** Yunxiang Long, Yingmu Tong, Yang Wu, Hai Wang, Chang Liu, Kai Qu, Guoliang Li

**Affiliations:** 1Department of Hepatobiliary Surgery, The First Affiliated Hospital of Xi’an Jiaotong University, Xi’an 710061, China; 2Department of General Surgery, The First Affiliated Hospital of Xi’an Jiaotong University, Xi’an 710061, China; 3Department of SICU, The First Affiliated Hospital of Xi’an Jiaotong University, Xi’an 710061, China; 4Department of Cardiovascular Medicine, The First Affiliated Hospital of Xi’an Jiaotong University, Xi’an 710061, China

**Keywords:** acute myocardial infarction, acute hepatic injury, coagulation dysfunction, prognosis

## Abstract

Background: In acute myocardial infarction (AMI), acute hepatic injury is an independent risk factor for prognosis and is associated with complex coagulation dynamics. This study aims to determine the interaction between acute hepatic injury and coagulation dysfunction on outcomes in AMI patients. Methods: The Medical Information Mart for Intensive Care (MIMIC-III) database was used to identify AMI patients who underwent liver function testing within 24 h of admission. After ruling out previous hepatic injury, patients were divided into the hepatic injury group and the nonhepatic injury group based on whether the alanine transaminase (ALT) level at admission was >3 times the upper limit of normal (ULN). The primary outcome was intensive care unit (ICU) mortality. Results: Among 703 AMI patients (67.994% male, median age 65.139 years (55.757–76.859)), acute hepatic injury occurred in 15.220% (*n* = 107). Compared with the nonhepatic injury group, patients with hepatic injury had a higher Elixhauser comorbidity index (ECI) score (12 (6–18) vs. 7 (1–12), *p* < 0.001) and more severe coagulation dysfunction (85.047% vs. 68.960%, *p* < 0.001). In addition, acute hepatic injury was associated with increased in-hospital mortality (odds ratio (OR) = 3.906; 95% CI: 2.053–7.433; *p* < 0.001), ICU mortality (OR = 4.866; 95% CI: 2.489–9.514; *p* < 0.001), 28-day mortality (OR = 4.129; 95% CI: 2.215–7.695; *p* < 0.001) and 90-day mortality (OR = 3.407; 95% CI: 1.883–6.165; *p* < 0.001) only in patients with coagulation disorder but not with normal coagulation. Unlike patients with coagulation disorder and normal liver, patients with both coagulation disorder and acute hepatic injury had greater odds of ICU mortality (OR = 8.565; 95% CI: 3.467–21.160; *p* < 0.001) than those with normal coagulation. Conclusions: The effects of acute hepatic injury on prognosis are likely to be modulated by early coagulation disorder in AMI patients.

## 1. Introduction

AMI is one of the most serious manifestations of coronary artery disease [1]. Emerging strategies for diagnosis, treatment and secondary prevention have improved the prognosis of AMI patients [2]; however, several adverse effects of extracardiac organ dysfunction that blunt the benefits of anti-AMI therapies have gradually attracted much attention [3].

The liver is highly vascular and metabolically active, accounting for approximately a quarter of the cardiac output, and it is sensitive to hemodynamic changes. Although the liver is capable of resisting minor circulatory disturbances by virtue of the dual blood supply from the portal vein and hepatic artery [4], AMI patients often present abnormal liver function, which is frequently observed in the first blood test for emergency medical admissions, especially in patients without any identifiable cause of hepatic injury. Previous studies have shown that hepatic functional changes on admission can effectively predict the prognosis of AMI patients [5]. In addition, an impaired hepatic function is associated with a hypercoagulable state and hemorrhagic complications due to the pathological imbalance between procoagulant and anticoagulant plasma factors [6]. The changes in coagulation function are rather complicated when AMI patients experience acute hepatic injury. However, the effect of acute hepatic injury and coagulation disorder on prognosis is unknown. Therefore, this study aims to explore the interaction between acute hepatic injury and coagulation disorder on the mortality of AMI patients.

## 2. Materials and Methods

### 2.1. Design and Setting

Data contained within the retrospective cohort study were retrieved from the MIMIC III database, a freely available database published by the Massachusetts Institute of Technology (MIT). The MIMIC-III database consists of 53,423 hospital admissions for adult patients who were admitted to intensive care units between 2001 and 2012. Data were not available until approval for access was granted by MIT’s institutional review board. All data were extracted by an author (Record ID: 28572693) who completed the CITI “Data or Specimens Only Research” course and passed the exam.

### 2.2. Study Population

All participants were adult patients (age ≥ 18 years) who were diagnosed with AMI and underwent liver function testing within 24 h of admission. The exclusion criteria were as follows: (1) pregnancy, (2) congenital coagulation disorders (e.g., hemophilia, von Willebrand disease, and vitamin K deficiency), (3) malignant tumors, (4) various causes of previous hepatic injury (hepatitis virus carriers, fatty liver, hepatitis, cirrhosis, parasitic liver disease, hepatobiliary obstructive disease, and other liver diseases), and (5) admission to ICU for AMI patients with primary cardiac diseases, such as congenital heart disease and valvular heart disease.

Patients were divided into the hepatic injury group and the nonhepatic injury group according to whether the alanine transaminase (ALT) level was >3 ULN on admission.

### 2.3. Definition and Outcomes

AMI should be considered when myocardial injury and necrosis occur and are accompanied by the clinical condition of myocardial ischemia. Myocardial injury was defined as at least one cardiac troponin value above the 99th percentile reference upper limit [2]. Acute hepatic injury was defined as ALT > 3 ULN on the first test of liver function on admission, without known cause for previous hepatic injury. The cutoff value of ALT was consistent with grade 2 to 5 treatment-related adverse events (AEs) (CTCAE v 5.0) [7]. ALT can specifically reflect liver dysfunction, as elevated aspartate aminotransferase (AST) has been shown in the case of ischemia-cell death of other tissues, including the kidney, skeletal muscle and brain. Given the greater specificity of ALT than AST, ALT was eventually chosen to evaluate acute hepatic injury [8]. Coagulation disorder within 24 h of admission was assessed according to one of three major indicators, platelets (PLT) < 1.50 × 10^11^/L, activated partial thromboplastin time (APTT) > 39 s or international normalized ratio (INR) > 1.4.

The primary outcome was ICU mortality. The secondary outcomes were in-hospital mortality, 28-day mortality and 90-day mortality.

### 2.4. Measurements

The study variables included demographics (age, gender, body mass index (BMI), ethnicity, insurance, marital status, unit type), complications (ECI, hypertension, diabetes, chronic pulmonary disease (CPD), peripheral vascular disease (PVD), atrial fibrillation (AF), ventricular fibrillation (VF), congestive heart failure (CHF), cardiogenic shock (CS) and cardiac arrest (CA)), clinical variables within 24 h of admission (peak level of creatine kinase-MB (CK-MB), creatine kinase (CK), lactate dehydrogenase (LDH), AST, alkaline phosphatase (ALP), INR, and APTT, the minimum value of PLT and albumin (ALB)), and hospitalized treatments (percutaneous coronary intervention (PCI), coronary artery bypass grafting (CABG), thrombolysis, β-blocker, statins, anticoagulation, antiplatelet and dual antiplatelet therapy (DAPT)).

### 2.5. Statistical Analysis

AMI patients were stratified by their baseline ALT values (hepatic injury, >3 ULN; nonhepatic injury, ≤3 ULN).Variables were summarized and compared between the hepatic injury group and the nonhepatic injury group. Continuous variables were reported as medians and ranges and compared with the Wilcoxon rank sum test. In addition, categorical variables were described as frequencies and percentages and compared with the chi-square test. Univariate and multivariate logistic regression analyses were used to test the association between acute hepatic injury and coagulation disorder on mortality. The confounders included age, gender, log_2_CK-MB, ECI, anticoagulation, antiplatelet, PCI, CABG, thrombolysis and coagulation disorder. Data analyses were performed using the SPSS (version 26.0 package (IBM, Armonk, NY, USA). Figures were plotted using GraphPad Prism 8.0 (GraphPad Prism Software Inc., San Diego, CA, USA). *p* < 0.05 was considered statistically significant.

## 3. Results

### 3.1. Characteristics of the Study Cohort

Ultimately, this study enrolled 703 subjects, including 478 (67.994%) male patients and 107 patients (15.220%) with acute hepatic injury (Figure 1). In the whole cohort, the median age was 65.139 years (IQR: 55.757–76.859). Compared with AMI patients in the nonhepatic injury group, patients in the hepatic injury group had a higher ECI (12 (6–18) vs. 7 (1–12), *p* < 0.001). Besides, a larger proportion of participants developed ventricular fibrillation (28 (26.168%) vs. 41 (6.879%), *p* < 0.001), cardiogenic shock (53 (49.533%) vs. 110 (18.456%), *p* < 0.001) and cardiac arrest (36 (33.645%) vs. 45 (7.550%), *p* < 0.001) in the hepatic injury group than in the nonhepatic injury group. However, there were no differences between two groups for anti-AMI therapies, such as PCI, CABG and antiplatelet therapy. Moreover, patients with acute hepatic injury showed higher in-hospital mortality (38 (35.514%) vs. 61 (10.235%), *p* < 0.001), ICU mortality (37 (34.579%) vs. 49 (8.221%), *p* < 0.001), 28-day mortality (40 (37.383%) vs. 65 (10.906%), *p* < 0.001) and 90-day mortality (46 (42.991%) vs. 87 (14.597%), *p* < 0.001) than patients without acute hepatic injury (Table 1 and Figure 2).

### 3.2. The Correlation between Acute Hepatic Injury and Laboratory Test Indices

AMI patients with hepatic injury had worse coagulation, cardiac and hepatic functions than patients without hepatic injury (Appendix A). A higher percentage of the population had abnormal INR (54 (52.941%) vs. 153 (27.224%), *p* < 0.001) and APTT values (83 (79.808%) vs. 355 (63.167%), *p* = 0.001) in the hepatic injury group. In addition, there were more patients presenting coagulation disorder (91 (85.047%) vs. 411 (68.960%), *p* < 0.001) in the hepatic injury group than in the nonhepatic injury group.

Furthermore, compared to patients with single acute hepatic injury or coagulation disorder, subjects with acute hepatic injury and abnormal coagulation function had more severe adverse outcomes, including in-hospital mortality, ICU mortality, 28-day mortality and 90-day mortality (Appendix A).

### 3.3. Association between Acute Hepatic Injury and Mortality

Multivariate logistic regression analysis (Table 2) showed that acute hepatic injury was independently associated with increased in-hospital mortality (OR = 3.387; 95% CI: 1.863–6.158; *p* < 0.001), ICU mortality (OR = 4.131; 95% CI: 2.227–7.662; *p* < 0.001), 28-day mortality (OR = 3.538; 95% CI: 1.977–6.332; *p* < 0.001) and 90-day mortality (OR = 3.282; 95% CI: 1.887–5.709; *p* < 0.001). More importantly, acute hepatic injury presented a significant association with mortality in AMI patients with coagulation dysfunction. In the maximally adjusted analysis, patients with coagulation disorder in the hepatic injury group were 290.6% (OR = 3.906; 95% CI: 2.053–7.433; *p* < 0.001), 386.6% (OR = 4.866; 95% CI: 2.489–9.514; *p* < 0.001), 312.9% (OR = 4.129; 95% CI: 2.215–7.695; *p* < 0.001) and 240.7% (OR = 3.407; 95% CI: 1.883–6.165; *p* < 0.001) more likely to experience in-hospital mortality, ICU mortality, 28-day mortality and 90-day mortality than those in the nonhepatic injury group, respectively. Nevertheless, no significant relationship between acute hepatic injury and mortality was found in AMI patients with normal coagulation function (Table 3).

### 3.4. Sensitivity Analysis of the Relationship between Acute Hepatic Injury and Mortality

In the population undergoing PCI, acute hepatic injury was still an independent predictor for increased in-hospital mortality (OR = 3.812; 95% CI: 1.784–8.149; *p* < 0.001), ICU mortality (OR = 5.058; 95% CI: 2.281–11.220; *p* < 0.001), 28-day mortality (OR = 3.856; 95% CI: 1.809–8.220; *p* < 0.001) and 90-day mortality (OR = 4.001; 95% CI: 1.931–8.290; *p* < 0.001). Among patients with coagulation disorder and PCI, participants with acute hepatic injury had a 324.3% (OR = 4.243; 95% CI: 1.891–9.522; *p* < 0.001), 493.3% (OR = 5.933; 95% CI: 2.487–14.153; *p* < 0.001), 322.8% (OR = 4.228; 95% CI:1.893–9.445; *p* < 0.001) and 292.1% (OR = 3.921; 95% CI: 1.801–8.537; *p* < 0.001) increased odds of in-hospital mortality, ICU mortality, 28-day mortality and 90-day mortality than those without acute hepatic injury, respectively. However, there was no association in the participants with normal coagulation function (Appendix A).

Similarly, among the AMI patients being administered anticoagulant or antiplatelet therapies, only the participants with coagulation disorder presented associations between acute hepatic injury and in-hospital mortality (OR = 5.209; 95% CI: 2.373–11.434; *p* < 0.001), ICU mortality (OR = 6.350; 95% CI: 2.736–14.739; *p* < 0.001), 28-day mortality (OR = 5.390; 95% CI: 2.516–11.547; *p* < 0.001) and 90-day mortality (OR = 4.211; 95% CI: 2.804–8.507; *p* < 0.001) (Appendix A).

### 3.5. The Joint Effects of Acute Hepatic Injury and Coagulation Disorder on Mortality

The significant hepatic injury × coagulation disorder interaction was examined by comparing coagulation disorder to normal coagulation function within the same type of liver function. In AMI patients with acute hepatic injury, patients with coagulation disorder had higher in-hospital mortality (OR = 15.918; 95% CI: 1.384–183.140; *p* = 0.026), ICU mortality (OR = 12.124; 95% CI: 1.138–129.205; *p* = 0.039) and 28-day mortality (OR = 13.697; 95% CI: 1.432–131.023; *p* = 0.023) than those with normal coagulation. However, the association between coagulation disorder and outcomes was not significant in AMI patients without hepatic injury (Appendix A).

Additionally, significantly increased odds of mortality were found among the patients with both coagulation disorder and acute hepatic injury. Compared with AMI patients with normal coagulation, greater odds of in-hospital mortality (OR = 8.517; 95% CI: 3.537–20.509; *p* < 0.001), ICU mortality (OR = 8.565; 95% CI: 3.467–21.160; *p* < 0.001) and 28-day mortality (OR = 8.869; 95% CI: 3.809–20.655; *p* < 0.001) were observed in patients with coagulation disorder and acute hepatic injury, while the odds ratios were not significant in the patients with coagulation disorder, but no acute hepatic injury (Table 4).

## 4. Discussion

In this study, we explored the relationship between acute hepatic injury and prognosis in AMI patients with normal coagulation function and coagulation disorder within 24 h of admission. AMI patients with acute hepatic injury had higher APTT and INR values, as well as a greater percentage of coagulation disorder than those without hepatic injury. In addition, acute hepatic injury was an independent risk factor for increased in-hospital mortality, ICU mortality, 28-day mortality and 90-day mortality. Furthermore, our results indicated a striking interaction between acute hepatic injury and coagulation disorder on the prognosis of AMI patients. The patients with acute hepatic injury exhibited higher odds of mortality than those without hepatic injury only when AMI patients presented coagulation disorder. Therefore, the incorporation of acute hepatic injury and coagulation disorder, which have not been included in existing models [3,9,10], would improve risk stratification for mortality.

Serum transaminase levels at admission are closely related to in-hospital mortality and long-term mortality of AMI patients [5,11,12]. We defined acute hepatic injury by ALT in consideration of its specificity since ALT is primarily produced by hepatocytes, whereas AST is a mitochondrial enzyme not only in the liver but also in skeletal muscle, cardiac muscle, kidney and brain tissue [13]. Our study reveals that acute hepatic injury is associated with increased mortality, which is consistent with previous research [5]. This phenomenon could be partially explained by the existence of heart-liver crosstalk. Bannon and colleagues revealed that the more severe cardiac insufficiency, the more rapidly hepatic enzymes were elevated. For example, mean levels of maximum ALT, first and maximum ALP and first and maximum gamma-glutamyl transpeptidase (GGT) values would increase as a gradient in the case of myocardial infarction [14]. In addition, in an AMI model of porcine closed chest reperfusion, myocardial infarction changed the expression of 856 hepatic genes (519 upregulated and 337 downregulated) in the liver [15]. The upregulated genes were closely related to metabolism, inflammation, cancer-related pathways, MAPK and chemokine signaling pathways, and the downregulated genes were related to the PPAR signaling pathway, biosynthesis of unsaturated fatty acids, fibrosis and extracellular matrix (ECM) receptor interaction. Moreover, the liver had the highest variation in reported metabolites, which was attributed to its role as one of the most metabolically active tissues [16]. Therefore, the effects of myocardial ischemia are not confined to the damaged myocardium, and multiple organs are also involved in the response to MI.

The specific mechanism of heart-liver crosstalk in AMI is unclear. On the one hand, the hepatic injury is derived from hypoperfusion secondary to AMI. It is well known that the liver receives a dual blood supply. Increased hepatic arterial blood flow can buffer 25% to 60% of the decreased flow in the portal vein [17]. However, the liver is still very sensitive to hemodynamic changes because of its complex vascular system, active metabolism and high perfusion rate [18]. Acute circulatory changes, such as AMI-associated hypotension, directly affect liver blood flow, thereby increasing ALT and AST. Even hypoperfusion-induced cell damage can be further exacerbated after blood perfusion is restored. However, additional hepatic congestion also promotes the development of acute cardiogenic hepatic injury [17]. Acute cardiogenic hepatic injury is usually associated with acute coronary events, arrhythmias, or temporary severe hypotension [19]. During cardiogenic shock, a sharp decline in hepatic perfusion is not the only cause of acute hepatic injury; the latter also results from hepatic venous congestion caused by right heart failure [20]. Therefore, acute cardiogenic hepatic injury results from a combination of hypoperfusion and hepatic hyperemia.

In this study, more than half of AMI patients had coagulation dysfunction at admission. Myocardial infarction is considered to be the result of a ‘perfect storm’ scenario in which the overlap of vulnerable plaques and thrombogenic blood is a key determinant of myocardial infarction occurrence and extension [21]. Generally, the mechanism of AMI initiates the rupture or erosion of fragile, lipid-rich atherosclerotic coronary plaques, which generates an imbalance in the coagulation and fibrinolysis systems [22]. When circulating blood is exposed to a highly thrombotic plaque core and stromal materials, platelets are stimulated to initiate adhesion, activation and aggregation, causing thrombin generation; this further accelerates platelet activation, forming fibrin effusion to capture red blood cells; eventually, thrombosis occurs [23]. Finally, completely occluded clots usually cause ST-segment elevation myocardial infarction (STEMI), as partial occlusion or occlusion with collateral circulation results in non-ST segment elevation myocardial infarction (NSTEMI) or unstable angina (UA). Unfortunately, the hypercoagulable state triggered by vulnerable plaques can further lead to the rupture of additional susceptible atherosclerotic plaques, so there may be more than one culprit lesion in AMI patients [24].

Continuous coagulation activation and thrombin-mediated platelet activation play key roles in AMI thrombosis. In the acute stage of myocardial infarction, plasma fibrinopeptide A (FPA), cross-linked fibrin, platelet factor 4 and fibrinogen degradation products increase [25]. Furthermore, measurement of coagulation and fibrinolytic biomarkers contribute to the risk stratification for MI secondary prevention, such as prothrombin fragment 1 + 2 (F1 + F2), fibrinopeptide A (FPA), thrombus precursor protein (TpP) and D-D dimer [26]. Importantly, persistent hypercoagulability still appears for six months after AMI. Frey et al. reported highly dynamic variations in blood coagulation factor XIII (FXIII) during the progression of AMI, and the reduction of FXIIIa in patients during early myocardial infarction indicated a large area of myocardial infarction and low left ventricular ejection fraction after 1 year [27]. In addition, D-D dimer values during 2–3 weeks after discharge are positively correlated with the risk of recurrent myocardial infarction, stroke, and clinically relevant bleeding events at 2 years after AMI [28].

Our study showed an interaction between acute hepatic injury and coagulation disorder on mortality, which arises from the adverse effects of AMI and hepatic injury on coagulation function. In many diseases, the activation of hepatic coagulation system is an inevitable result of inflammatory cell activation and tissue damage. The liver, a main synthesis site of multiple coagulation factors and their inhibitors, plays a key role in thrombosis, hemostasis and inflammation. Thus, liver abnormalities are associated with a hypercoagulable state and hemorrhagic diseases [6]. Nonetheless, it is rare to observe bleeding complications in patients with acute liver failure (ALF) [29]. There is coagulation rebalancing occurring in ALF, that is, apparent hemostasis insufficiency in ALF can be rebalanced by compensatory mechanisms caused by systemic inflammation and endothelial cell activation or injury [30].

As a multicenter study, this study adds generalizability to the conclusions due to the heterogeneity of population. In addition, we are the first to evaluate the interaction of acute hepatic injury and coagulation dysfunction on patient mortality. Therefore, this work provides an important reference for individualized medicine, precise treatment and timely monitoring in AMI patients complicated with acute hepatic injury.

Nevertheless, there are also several limitations that warrant discussion. First, we only describe associations due to the natural limitations of retrospective studies. Second, there is a lack of fibrinolytic system indices, such as fibrinogen (FIB), D-D dimer, and fibrinogen degradation products (FDP) in our database. Only coagulation indicators, including APTT, PT and INR, are included. Third, middle-aged and elderly people are still the main AMI population, and the median age of the whole cohort was 65.139 years in this study. Therefore, our conclusion should be cautiously applied to young AMI individuals.

## 5. Conclusions

Coagulation within 24 h of admission can effectively stratify the mortality of AMI with acute hepatic injury. Acute hepatic injury was only associated with increased mortality in AMI patients with coagulation disorder.

## Figures and Tables

**Figure 1 jcm-12-01534-f001:**
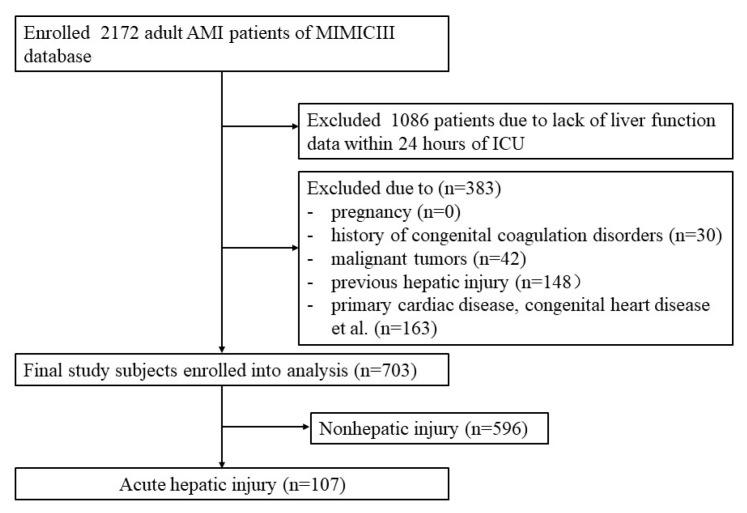
Flowchart of the enrolled patients.

**Figure 2 jcm-12-01534-f002:**
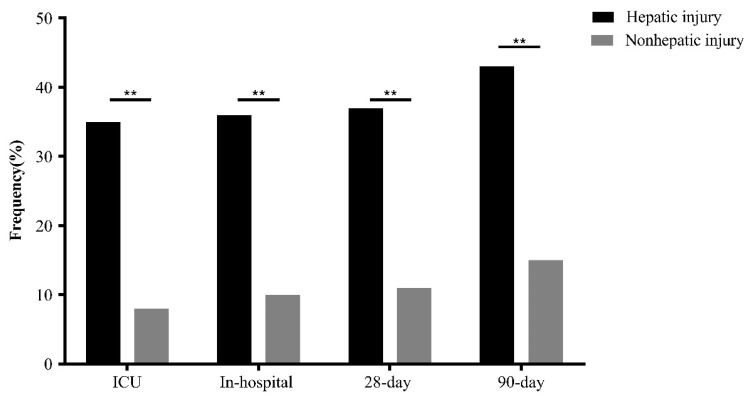
The mortality of AMI patients stratified by acute hepatic injury. **: *p* < 0.001.

**Table 1 jcm-12-01534-t001:** Demographic and clinical characteristics of AMI patients stratified by acute hepatic injury.

Variables	Total (*n* = 703)	Nonhepatic Injury (*n* = 596)	Hepatic Injury (*n* = 107)	*p* Value
Age	65.139 (55.757–76.859)	65.339 (55.757–76.695)	64.802 (53.016–79.193)	0.933
Gender (Male, %)	478 (67.994)	406 (68.121)	72 (67.290)	0.865
BMI	27.566 (24.316–31.026)	27.490 (24.277–31.026)	28.196 (24.802–30.831)	0.591
Ethnicity (*n*, %)				0.660
Asian	8 (1.538)	7 (1.556)	1 (1.429)	
Blank	37 (7.115)	31 (6.889)	6 (8.571)	
Hispanic	18 (3.462)	15 (3.333)	3 (4.286)	
White	436 (83.846)	379 (84.222)	57 (81.429)	
other	21 (4.038)	18 (4.000)	3 (4.286)	
Insurance (*n*, %)				0.431
Government	26 (3.698)	21 (3.523)	5 (4.673)	
Medicaid	34 (4.836)	28 (4.698)	6 (5.607)	
Medicare	356 (50.640)	300 (50.336)	56 (52.336)	
Private	276 (39.260)	239 (40.101)	37 (34.579)	
Self-Pay	11 (1.565)	8 (1.342)	3 (2.804)	
Marital status (*n*, %)				0.841
Divorced	54 (8.333)	47 (8.561)	7 (7.071)	
Married	380 (58.642)	320 (58.288)	60 (60.606)	
Separated	2 (0.309)	2 (0.364)	0 (0)	
Single	119 (18.364)	103 (18.761)	16 (16.162)	
Widowed	93 (14.352)	77 (14.026)	16 (16.162)	
Unit type (*n*, %)				0.513
CSICU	86 (12.268)	73 (12.290)	13 (12.150)	
CCU	519 (74.037)	437 (73.569)	82 (76.636)	
MICU	73 (10.414)	66 (11.111)	7 (6.542)	
SICU	13 (1.854)	11 (1.852)	2 (1.869)	
TSICU	10 (1.427)	7 (1.178)	3 (2.804)	
Comorbidities				
ECI	7 (3–12)	7 (1–12)	12 (6–18)	<0.001
Hypertension (*n*, %)	399 (56.757)	346 (58.054)	53 (49.533)	0.101
Diabetes (*n*, %)	191 (27.169)	163 (27.349)	28 (26.168)	0.800
CPD (*n*, %)	121 (17.212)	106 (17.785)	15 (14.019)	0.342
PVD (*n*, %)	56 (7.966)	50 (8.389)	6 (5.607)	0.328
AF (*n*, %)	157 (22.333)	128 (21.477)	29 (27.103)	0.198
VF (*n*, %)	69 (9.815)	41 (6.879)	28 (26.168)	<0.001
CHF (*n*, %)	275 (39.118)	227 (38.087)	48 (44.860)	0.186
CS (*n*, %)	163 (23.186)	110 (18.456)	53 (49.533)	<0.001
CA (*n*, %)	81 (11.522)	45 (7.550)	36 (33.645)	<0.001
Therapy (*n*, %)				
PCI	407 (59.329)	346 (59.450)	61 (58.654)	0.879
CABG	124 (18.076)	109 (18.729)	15 (14.423)	0.293
Thrombolysis	25 (3.644)	22 (3.780)	3 (2.885)	0.654
β-blocker	340 (48.364)	307 (51.510)	33 (30.841)	<0.001
Statins	337 (47.937)	292 (48.993)	45 (42.056)	0.186
Anticoagulation	305 (43.385)	248 (41.611)	57 (53.271)	0.025
DAPT	111 (15.789)	90 (15.100)	21 (19.626)	0.237
Antiplatelet	407 (57.895)	336 (56.376)	71 (66.355)	0.054
Length of hospital stay (Day)	5.546 (3.382–9.929)	5.358 (3.382–9.788)	6.763 (3.594–15.891)	0.109
ICU stay (Day)	2.586 (1.428–5.083)	2.434 (1.425–4.864)	3.585 (1.713–7.492)	0.001
Outcomes (*n*, %)				
In-hospital mortality	99 (14.083)	61 (10.235)	38 (35.514)	<0.001
ICU mortality	86 (12.233)	49 (8.221)	37 (34.579)	<0.001
28-day mortality	105 (14.936)	65 (10.906)	40 (37.383)	<0.001
90-day mortality	133 (18.919)	87 (14.597)	46 (42.991)	<0.001

**Table 2 jcm-12-01534-t002:** The association between acute hepatic injury and outcomes in AMI patients.

	OR; 95% CI	*p* Value
Model 1		
In-hospital mortality	4.830 (3.000–7.777)	<0.001
ICU mortality	5.901 (3.600–9.671)	<0.001
28-day mortality	4.877 (3.052–7.793)	<0.001
90-day mortality	4.412 (2.827–6.886)	<0.001
Model 2		
In-hospital mortality	4.981 (3.056–8.118)	<0.001
ICU mortality	6.141 (3.695–10.207)	<0.001
28-day mortality	5.151 (3.167–8.376)	<0.001
90-day mortality	4.605 (2.910–7.285)	<0.001
Model 3		
In-hospital mortality	3.677 (2.027–6.672)	<0.001
ICU mortality	4.486 (2.425–8.298)	<0.001
28-day mortality	3.837 (2.148–6.856)	<0.001
90-day mortality	3.596 (2.071–6.242)	<0.001
Model 4		
In-hospital mortality	3.387 (1.863–6.158)	<0.001
ICU mortality	4.131 (2.227–7.662)	<0.001
28-day mortality	3.538 (1.977–6.332)	<0.001
90-day mortality	3.282 (1.887–5.709)	<0.001

Model 1: univariate analysis; Model 2: Covariates of multivariate regression, including age, gender; Model 3: Covariates of multivariate regression, including age, gender, log_2_CKMB, ECI, anticoagulation, antiplatelet, PCI, CABG, thrombolysis; Model 4: Covariates of multivariate regression, including age, gender, log_2_CKMB, ECI, anticoagulation, antiplatelet, PCI, CABG, thrombolysis, coagulation disorder.

**Table 3 jcm-12-01534-t003:** Multivariate Regression of acute hepatic injury related to outcomes stratified by coagulation function.

Predictors	Univariate Regression Analysis		Multivariate Regression Analysis	
	OR; 95% CI	*p* Value	OR; 95% CI	*p* Value
Coagulation disorder				
Nonhepatic injury	reference		reference	
In-hospital mortality	4.947 (2.964–8.257)	<0.001	3.906 (2.053–7.433)	<0.001
ICU mortality	6.071 (3.567–10.334)	<0.001	4.866 (2.489–9.514)	<0.001
28-day mortality	5.066 (3.056–8.399)	<0.001	4.129 (2.215–7.695)	<0.001
90-day mortality	4.013 (2.479–6.497)	<0.001	3.407 (1.883–6.165)	<0.001
Normal coagulation function				
Nonhepatic injury	reference		reference	
In-hospital mortality	1.055 (0.127–8.733)	0.961	0.439 (0.021–9.308)	0.597
ICU mortality	1.304 (0.155–10.994)	0.807	0.447 (0.008–23.806)	0.691
28-day mortality	0.961 (0.117–7.905)	0.971	0.317 (0.012–8.200)	0.489
90-day mortality	3.327 (0.833–13.291)	0.089	3.075 (0.480–19.714)	0.236

Covariates of multivariate regression, including age, gender, log_2_CKMB, ECI, anticoagulation, antiplatelet, PCI, CABG and thrombolysis.

**Table 4 jcm-12-01534-t004:** The joint effects of acute hepatic injury and coagulation disorder on mortality.

Predictors	Univariate Regression Analysis		Multivariate Regression Analysis	
	OR; 95% CI	*p* Value	OR; 95% CI	*p* Value
Normal coagulation function	reference		reference	
Abnormal coagulation-liver function				
Nonhepatic injury				
In-hospital mortality	1.919 (1.035–3.560)	0.039	1.781 (0.864–3.671)	0.118
ICU mortality	1.761 (0.903–3.434)	0.097	1.695 (0.777–3.695)	0.185
28-day mortality	1.905 (1.046–3.469)	0.035	1.713 (0.846–3.470)	0.135
90-day mortality	2.508 (1.439–4.371)	0.001	2.204 (1.151–4.217)	0.017
Hepatic injury				
In-hospital mortality	9.495 (4.786–18.834)	<0.001	8.517 (3.537–20.509)	<0.001
ICU mortality	10.691 (5.212–21.931)	<0.001	8.565 (3.467–21.160)	<0.001
28-day mortality	9.650 (4.940–18.851)	<0.001	8.869 (3.809–20.655)	<0.001
90-day mortality	10.065 (5.283–19.175)	<0.001	8.389 (3.742–18.811)	<0.001

Covariates of multivariate regression, including age, gender, log_2_CKMB, ECI, anticoagulation, antiplatelet, PCI, CABG and thrombolysis.

## Data Availability

No new data were created.

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
