# Peer review of "Interaction between Acute Hepatic Injury and Early Coagulation Dysfunction on Mortality in Patients with Acute Myocardial Infarction"

_jcm, 2023, doi:10.3390/jcm12041534_

Round 1
Reviewer 1 Report
The MS by Long et al. investigates the role of hepatic injury and coagulation abnormalities in a cohort of presumably AMI patients in a freely available database from the Massachusetts Institute of Technology (MIMIC III). They find that hepatic injury and coagulation abnormalities in AMI patients are linked to an increased Odd ratio for mortality. Organ dysfunction is a well defined risk marker for ICU mortality therefore the added value of this study is limited. In addition, coagulation abnormalities were assessed by rather rough measures including aPTT, INR and PLT counts and may just represent more sever hepatic injury. While the statistical approach appears to be valid the the manuscript is sometimes difficult to read and language support should be provided.
In addition I have a few remarks:
Major:
- Please clarify: Are these patients true AMI patients or patients with myocardial injury. For example chronic kidney disease will lead to an increase of Troponin which is not necessarily a results of myocardial injury. In addition, Type 2 myocardial infarction has a different cause that Type 1 myocardial infarction. Please clarify.
- Rather unspecific coagulation test were used. Patients with AMI will receive anticoagulation during PCI and depending on the outcome even after PCI. How did you correct for this?
- The authors imply an interaction between coagulation abnormalities and liver injury which is well established. Many coagulation factors are synthesised in the liver and will therefore directly affect coagulation. Therefore, patients with a coagulation abnormality and hepatic injury most likely represent patients with more sever liver injury. Therefore a formal interaction analysis should be performed were the interaciton between coagulation abnormalities and liver injury is tested and adjusted for the isolated effect of liver injury and coagulation abnormalities.
- If the authors wish to proof that liver injury and coagulation abnormalities add to current existing risk models they should test the additive value of these parameters on risk factors by a c statistic.
Minor:
- Please thoroughly recheck grammer and language.
- Please introduce acronyms before when these are used first time. At current many acronyms are use in the abstract without introduction (i.e. MIMIC-III; ECI Score etc. )
Author Response
- Please clarify: Are these patients true AMI patients or patients with myocardial injury. For example, chronic kidney disease will lead to an increase of Troponin which is not necessarily a result of myocardial injury. In addition, Type 2 myocardial infarction has a different cause that Type 1 myocardial infarction. Please clarify.
Response:
Thanks a lot for your comment. In this study, we selected the population whose discharge diagnosis included acute myocardial infarction with reference to ICD-9 diagnostic codes. Patients with single myocardial injury were excluded. In addition, because the MIMICIII database contains participants from multiple medical centers, it is difficult to determine primary or secondary myocardial infarction based on diagnosis in partial patients. So, we apologize for not being able to distinguish between myocardial infarction types.
In addition, we re-screened the patient population according to previous inclusion and exclusion criteria. Due to the existence of duplicate cases, we finally included a total of 703 patients and re-revised the results
- Rather unspecific coagulation test was used. Patients with AMI will receive anticoagulation during PCI and depending on the outcome even after PCI. How did you correct for this?
Response: Thanks a lot for your comment. First, to eliminate the effects of PCI and anticoagulation on outcomes, we used multiple logistic regression models to examine the association between acute hepatic injury and mortality, adjusting for age, gender, log2CKMB, ECI, SAPSII, anticoagulation, antiplatelet, PCI, CABG, thrombolysis and coagulation disorder in model 4 (Table 2). In addition, we also performed sensitivity analysis and found acute hepatic injury was independently associated with ICU mortality in PCI population after adjusting for anticoagulation treatment (supplementary table 3).
- The authors imply an interaction between coagulation abnormalities and liver injury which is well established. Many coagulation factors are synthesized in the liver and will therefore directly affect coagulation. Therefore, patients with a coagulation abnormality and hepatic injury most likely represent patients with more sever liver injury. Therefore, a formal interaction analysis should be performed were the interaction between coagulation abnormalities and liver injury is tested and adjusted for the isolated effect of liver injury and coagulation abnormalities.
Response: Thanks a lot for your comment. We have supplemented and revised the results to support the interaction between coagulation abnormalities and liver injury on mortality. First of all, the results showed that acute hepatic injury and coagulation disorder were independent factors of mortality in AMI patients (Table 2 and Supplementary table 4). In further analysis, the effects of acute hepatic injury on prognosis can be modulated by early coagulation disorder. Similarly, only among patients with acute hepatic injury, coagulation disorder was negatively correlated with the prognosis of patients (Table 2 and Supplementary table 4). Finally, significantly increased odds of mortality were found among the patients with coagulation disorder and acute hepatic injury, not patients with single coagulation disorder, than those with normal coagulation function (Table 4).
- If the authors wish to proof that liver injury and coagulation abnormalities add to current existing risk models, they should test the additive value of these parameters on risk factors by a c statistic.
Response: Thanks a lot for your comment. In this study, we simply focused on the interaction between coagulation disorder and acute hepatic injury on mortality. If we further proof the influence of hepatic injury and coagulation disorder on current existing risk models, this will cause the results of this article too complex. Therefore, we will continue to explore the optimal prognostic model of acute myocardial infarction which will consist of multiple variables, including acute hepatic injury and coagulopathy, in the following study.

Reviewer 2 Report
This Manuscript is an interesting analysis of the correlation between acute hepatic injury and coagulation dysfunction on outcomes in patients with acute myocardial infarction. The authors found that AMI patients with coagulation disorders and acute hepatic injury presented higher mortality compared with patients with normal coagulation or no acute hepatic injury.
The manuscript is quite clear and presented in a quite well-structured manner. I have only minor revisions:
1) The English should be a little revised and in my opinion some sentences should be rearranged (for example line 129: “The study was eventually of comprised 718 subjects, of whom 122 (16.992%) had acute hepatic injury” is not so clear)
2) The text should be re-read carefully because there are some typos (i.e. line 64 “53,423 hospital admissions for adult inpatients admitted to the intensive care units” or line 182 “resented” instead of presented)
3) Please verify the unit of length of platelets at line 92. Are you sure that are Litres?
Author Response
1. Please thoroughly recheck grammer and language.
Response: Thanks a lot for your comment. We polished the article again on professional article polishing site-AJE and checked the grammar and language sections carefully.
In addition, we re-screened the patient population according to previous inclusion and exclusion criteria. Due to the existence of duplicate cases, we finally included a total of 703 patients and re-revised the results.
- Please introduce acronyms before when these are used first time. At current many acronyms are use in the abstract without introduction (i.e., MIMIC-III; ECI Score etc.)
Response: Thanks a lot for your comment. We have rechecked the abbreviations to ensure that each abbreviation retains its full name when it first appears.
- The manuscript is quite clear and presented in a quite well-structured manner. I have only minor revisions:
1) The English should be a little revised and in my opinion some sentences should be rearranged (for example line 129: “The study was eventually of comprised 718 subjects, of whom 122 (16.992%) had acute hepatic injury” is not so clear)
Response: Done. We have revised this sentence. “Ultimately, this study enrolled 703 subjects, including 478 (67.994%) male patients and 107 patients (15.220%) with acute hepatic injury”
2) The text should be re-read carefully because there are some typos (i.e., line 64 “53,423 hospital admissions for adult inpatients admitted to the intensive care units” or line 182 “resented” instead of presented)
Response: Done
3) Please verify the unit of length of platelets at line 92. Are you sure that are Litres?
Response: Thanks a lot for your comment. We have verified that the platelet unit was 10^9/L in the clinic.

Round 2
Reviewer 1 Report
I have no further comments.